### Global fire emissions estimates during 1997-2015

Guido R. van der Werf<sup>1</sup>, James T. Randerson<sup>2</sup>, Louis Giglio<sup>3</sup>, Thijs T. van Leeuwen<sup>4,5</sup>, Yang Chen<sup>2</sup>, Brendan M. Rogers<sup>6</sup>, Mingquan Mu<sup>2</sup>, Margreet J.E. van Marle<sup>1</sup>, Douglas C. Morton<sup>7</sup>, G. James Collatz<sup>7</sup>, Robert J. Yokelson<sup>8</sup>, Prasad S. Kasibhatla<sup>9</sup>

- <sup>5</sup> <sup>1</sup>Faculty of Earth and Life Sciences, Vrije Universiteit Amsterdam, Amsterdam, 1081HV, the Netherlands <sup>2</sup>Department of Earth System Science, University of California, Irvine, CA, 92697, US <sup>3</sup>Department of Geographical Sciences, University of Maryland, MD, 20742, US <sup>4</sup>SRON Netherlands Institute for Space Research, Utrecht, 3584CA, the Netherlands <sup>5</sup>now at VanderSat BV, Noordwijk, 2201DK, the Netherlands
- <sup>6</sup>Woods Hole Research Center, Falmouth, MA, 02540, US
   <sup>7</sup>NASA Goddard Space Flight Center, Greenbelt, MD, 20771, US
   <sup>8</sup>Department of Chemistry, University of Montana, Missoula, MT, 59812, US
   <sup>9</sup>Nicholas School of the Environment, Duke University, Durham, NC, 27708, US

Correspondence to: Guido van der Werf (guido.vander.werf@vu.nl)

- Abstract. Climate, land use, and other anthropogenic and natural drivers have the potential to influence fire dynamics in many regions. To develop a mechanistic understanding of the changing role of these drivers and their impact on atmospheric composition, long term fire records are needed that fuse information from different satellite and in-situ data streams. Here we describe the fourth version of the Global Fire Emissions Database (GFED) and quantify global fire emissions patterns during 1997-2015. The modeling system, based on the
- Carnegie-Ames-Stanford-Approach (CASA) biogeochemical model, has several modifications from the previous version and uses higher quality input datasets. Significant upgrades include: 1) new burned area estimates with contributions from small fires, 2) a revised fuel consumption parameterization optimized using field observations, 3) modifications that improve the representation of fuel consumption in frequently burning landscapes, and 4) fire severity estimates that better represent continental differences in burning processes across
- boreal regions of North America and Eurasia. The new version has a higher spatial resolution (0.25°) and uses a different set of emission factors that separately resolves trace gas and aerosol emissions from temperate and boreal forest ecosystems. Global mean carbon emissions using the burned area dataset with small fires (GFED4s) were 2.2 × 10<sup>15</sup> grams carbon per year (Pg C yr<sup>-1</sup>) during 1997-2015, with a maximum in 1997 (3.0 Pg C yr<sup>-1</sup>) and minimum in 2013 (1.8 Pg C yr<sup>-1</sup>). These estimates were 11 % higher than our previous estimates
- (GFED3) during 1997-2011, when the two datasets overlapped. This increase was the result of a substantial increase in burned area (37 %), mostly due to the inclusion of small fires, and a modest decrease in mean fuel consumption (-19 %) to better match estimates from field studies, primarily in savannas and grasslands. For trace gas and aerosol emissions, differences between GFED4s and GFED3 were often larger due to the use of revised emission factors. If small fire burned area was excluded (GFED4 without the "s" for small fires), average
- emissions were 1.5 Pg C yr<sup>-1</sup>. The addition of small fires had the largest impact on emissions in temperate North America, Central America, Europe, and temperate Asia. Our improved dataset provides an internally consistent set of burned area and emissions that may contribute to a better understanding of multi-decadal changes in fire dynamics and their impact on the Earth System. GFED data is available from http://www.globalfiredata.org.

### 1 Introduction

Fires have occurred naturally since the rise of vascular plants on land over 400 million years ago (Scott and Glasspool, 2006), shaping biomes and influencing climate through modulation of the carbon cycle and emissions of greenhouse gases and aerosols (Edwards et al., 2010; Langmann et al., 2009; van Langevelde et al., 2003).

- During the Anthropocene, humans have become an increasingly important driver of fire occurrence (Bowman et al., 2011). Human activity has enhanced fire activity in locations such as deforestation zones, while fire suppression and conversion of fire-prone landscapes such as savannas to agriculture in Africa, or of fire-maintained open lands to closed-canopy forests in the eastern US has generally decreased fire activity (Andela and van der Werf, 2014; Bowman et al., 2009; Nowacki and Abrams, 2008). To study how climate influences
- fires at the global scale and, in turn, how fires influence the carbon cycle, air quality, and climate we have developed the Global Fire Emissions Database (GFED). The scientific community has used past releases of GFED for over a decade. GFED has been used by atmospheric and biogeochemical modeling groups as an input dataset to study the impact of fires on biogeochemical cycles (Chen et al., 2010; Schwietzke et al., 2016), atmospheric chemistry (Aouizerats et al.,
- 2015; Castellanos et al., 2014), and human health (Johnston et al., 2012; Marlier et al., 2013), in assessment reports of the Intergovernmental Panel on Climate Change (IPCC) to estimate the role of fire and deforestation in biogeochemical cycles (Ciais et al., 2013), in the National Oceanic and Atmospheric Administration (NOAA's) Carbontracker system (Peters et al., 2007), and in annual updates of the Global Carbon Project (Le Queré et al., 2015). GFED also serves as a benchmark for optimizing fire modules in dynamic global vegetation and Earth
- System models (Hantson et al., 2016), and for fire emissions estimates derived from fire radiative power (FRP), including the Global Fire Assimilation System (Kaiser et al., 2012). Finally, burned area from GFED has provided a means for building early warning systems of fire season severity (Chen et al., 2016). The first version of GFED was released in 2004 and has since undergone several revisions as improved burned area estimates became available. GFED2 was released after Giglio et al. (2006) improved on the mapping of
- burned area from active fire data. GFED3 was released when this conversion was no longer necessary because almost all burned area in the Moderate Resolution Imaging Spectroradiometer (MODIS) era had been mapped (Giglio et al., 2010), and the current version follows further improvements in the burned area algorithm (Giglio et al., 2013). Satellite burned area is the most important input dataset regulating the spatial and temporal pattern of emissions following the Seiler and Crutzen (1980) approach, and is complemented in GFED by a
- biogeochemical modeling framework that provides estimates of biomass in various carbon 'pools' including leaves, grasses, stems, coarse woody debris, and litter. These pools are combusted to different degrees during a fire depending on pool-specific parameters and environmental conditions that influence fuel moisture and the simulated burn depth in organic soils of boreal forests and peatlands.
- Over the past decade, a parallel line of research has made considerable progress in estimating emissions using direct observations of fire radiative power (FRP). When continuous observations are available or the FRP diurnal cycle can be modeled, FRP can be integrated over time turning FRP into fire radiative energy (FRE) which is directly related to fire emissions (Wooster, 2002). FRP-based methods can provide emissions estimates in near

# Science Scienc

real time (Darmenov and da Silva, 2015; Kaiser et al., 2012). Despite progress (Ichoku and Ellison, 2014; Schroeder et al., 2014a), there is still substantial uncertainty and some of these FRE approaches apply a scaling factor to match GFED. Comparisons between the 'classical' burned area approach and the FRP approach, or approaches based on active fire detections in general, have indicated there is considerable variability in the

- amount of burned area associated with an individual active fire detection, and thus, the two approaches do not always align (Giglio et al., 2006; Randerson et al., 2012). In general, direct mapping of burned area excels when fires are large, but has difficulty in detecting smaller fires, for example, in croplands and in other areas where many fires have a size below the 21 ha of an individual 500 m MODIS pixel. Combining both burned area and active fire data, Randerson et al. (2012) provided evidence that the total area burned by these relatively small
- fires is substantial at the global scale. Therefore, emission estimates based solely on active fires, including the Fire INventory from NCAR (Wiedinmyer et al., 2011), may better capture spatial and temporal variability in regions with many small fires than emission estimates based solely on burned area. However, approaches based solely on active fires usually do not account for spatial and temporal variability in burned area per active fire nor in variability in fuel consumption within biomes.
- In this paper we describe the emissions estimates associated with the GFED4 burned area product from Giglio et al. (2013), with or without additional burned area from small fires based on a revised version of the Randerson et al. (2012) small-fire estimation approach. The main focus of our analysis will be on the model version that includes small fires (GFED4s), while the emissions estimates based on burned area without small fires will be referred to as GFED4. We also used a recent meta-analysis (van Leeuwen et al., 2014) to constrain our modeled
- estimates of fuel consumption. Fuel consumption is the amount of biomass, coarse and fine litter, and soil organic matter consumed per unit area burned and is the product of fuel load and combustion completeness. Besides these two main improvements over earlier versions, we made a number of additional modifications including updated input datasets and the use of satellite-derived estimates of parameters governing fuel consumption and tree mortality in the boreal region (Rogers et al., 2015). In Sect. 2 we provide more detail on
- these input datasets, followed by a description of the modeling framework in Sect. 3. Results are given in Sect. 4 followed by a discussion in Sect. 5 that includes a description of the main differences with GFED3 and an assessment of the primary sources of uncertainty in estimating fire emissions. In the conclusions (Sect. 6) we summarize the main points of our analysis and describe several important directions for future work.

### 2 Input datasets

Our version of the CASA model described in Sect. 3 requires input datasets on vegetation characteristics, meteorology, and fire parameters. Most of these datasets are somewhat different from those used in previous versions of GFED, in part due to a need for shorter latency in our updates. We re-gridded all of the input datasets to 0.25° spatial resolution and a monthly temporal resolution. We took additional steps to create estimates of fire dynamics on daily and 3-hourly time steps.

### 2.1 Vegetation characteristics

In CASA, the fraction of absorbed photosynthetically active radiation (fAPAR) is used to estimate net primary production (NPP), fractional tree cover (FTC) is used in the allocation of NPP between living carbon pools, and land cover (LC) is used to set turnover rates for stems and leaves, applying emission factors, and for categorizing

fire carbon emissions.

We calculated fAPAR based on the Global Inventory Modeling and Mapping Studies (GIMMS) Normalized Difference Vegetation Index (NDVI) version 3g (Pinzon and Tucker, 2014) and relations established by Los et al. (2000). This dataset is derived from the Advanced Very High Resolution Radiometer (AVHRR) sensor flying on board several satellites. We capped fAPAR at 0.95, corresponding to an NDVI value of 0.9. Data were not

available for several remote islands, including Hawaii and Fiji, and we do not report emissions for these locations.

FTC was derived by aggregating the annual MODIS MOD44B vegetation continuous fields (250m, V051, Hansen et al., 2005) to 0.25°. In order to provide consistency over the full time period, we used the last year available (2013) and increased FTC in prior years using the fire-driven deforestation rates. These fire-driven

deforestation rates were based on the amount of burned area within tropical forests at an annual time step. We used land cover maps from the annual MODIS MCD12C1 land cover type product and University of Maryland (UMD) classification scheme (Friedl et al., 2010). The climate modeling grid (CMG, 0.05°) dataset was resampled to 0.25° based on the mode land cover type. This dataset was available for 2001-2012; data from 2001 were applied to earlier years in the time series, and 2012 land cover data were used for years after 2012.

### 20 2.2 Meteorological datasets

We now use air temperature (t2m), soil moisture (swvl), and solar radiation (ssrd) from the ERA-interim dataset (Dee et al., 2011) produced by the European Centre for Medium-Range Weather Forecasts (ECMWF). We calculated the monthly mean for all datasets and regridded the 0.75° dataset to our 0.25° resolution without interpolation.

- These datasets are somewhat different from inputs for earlier GFED versions but are now internally consistent. Interannual and seasonal variability was relatively similar to datasets previously used in GFED, and these variations have the largest impact on our calculations. The use of soil moisture is new; previously, we used a bucket model based on rainfall and potential evaporation to calculate the wetness of soils, a key input dataset for calculating heterotrophic respiration (R<sub>h</sub>) rates and combustion completeness (see Sect. 3). Soil moisture is now
- transformed to a soil moisture index (SMI) based on soil-type specific permanent wilting point (PWP) and field capacity (FC) values as described in <a href="http://www.ecmwf.int/en/forecasts/documentation-and-support/evolution-ifs/cycles/change-soil-hydrology-scheme-ifs-cycle">http://www.ecmwf.int/en/forecasts/documentation-and-support/evolution-ifs/cycles/change-soil-hydrology-scheme-ifs-cycle</a> and is capped at 1. This was done for all 4 different soil layers (0-7, 8-28, 29-100, 101-255 cm). The SMI for the 0-7 cm layer replaces the scalar used previously for combustion completeness, the average SMI of all layers was used in the allocation of assimilated carbon to
- above- and belowground pools (see Sect. 3), the average SMI of the top two layers was used to down regulate NPP in herbaceous vegetation in the light use efficiency model when moisture was limiting, whereas the average

of the top four layers were used for NPP in woody vegetation. We used the average SMI for the upper two layers to represent the influence of soil moisture on the abiotic scalar regulating rates of  $R_{\rm h}$ .

### 2.3 Fire processes

- We derived burned area (both mapped burned area and active fire detections scaled to burned area) and metrics that can be used to assess fire-induced tree mortality and combustion completeness from satellite. Our burned area time series is based on MODIS data for the August 2000 onwards period (the "MODIS era") and based on other sensors before that period. In Sect. 2.3.1 we briefly describe the MODIS burned area data for which a more detailed description, including how the pre-MODIS burned area was derived, is described in Giglio et al. (2013).
- In Sect. 2.3.2 we then explain how the small fire burned area estimates for the MODIS era were derived based on Randerson et al. (2012). This is the GFED4s burned area time series and complemented with other sensors to compute the full 1997-2015 time period dataset (Sect. 2.3.3).

### 2.3.1 Burned area from MODIS

For the MODIS era we used the MODIS Collection 5.1 MCD64A1 burned area product (Giglio et al., 2013). Compared with Collection 5 and earlier versions of the MCD64A1, the Collection 5.1 product reduces the unintentional removal of small burns and eliminates some systematic omission errors (Giglio et al., 2013). The MCD64A1 product maps daily burned area at 500 m spatial resolution; these data are then aggregated to a 0.25° grid (both monthly and daily) to produce the MODIS-era GFED4 burned area product (Fig. 1a).

### 20

### 2.3.2 Small fire burned area during the MODIS era

In the MODIS era, we combined 500 m burned area (see above), 1-km thermal anomalies (active fires) from Terra and Aqua MODIS, and 500 m surface reflectance observations to statistically estimate burned area associated with small fires,  $BA_{sf}$ , in each 0.25° grid cell (*i*), month (*t*), and aggregated vegetation type ( $\nu$ ):

$$BA_{sf}(i,t,v) = FC_{out}(i,t,v) \times \alpha_{r,s,v,y} \times \gamma_{r,s,v,y}$$
(1)

where  $FC_{out}$  is the number of active fire pixels outside of the perimeter of the MCD64A1 burned area,  $\alpha$  is a ratio of burned area to active fires within MCD64A1 burned areas, and  $\gamma$  is a correction factor derived from comparing difference normalized burned area (dNBR) of active fires observed outside ( $dNBR_{out}$ ) and inside ( $dNBR_{in}$ ) of MCD64A1 burned areas with unburned control areas ( $dNBR_{control}$ , see Eq. 4 of Randerson et al., (2012)).  $\alpha$  and  $\gamma$  scalars were estimated each year ( $\gamma$ ), as a function of region (r), seasonal interval (s), and aggregated vegetation type ( $\nu$ ). Our method was similar to that described in Randerson et al. (2012), but with several important modifications to each of the 3 factors on the right hand side of Eq. 1 as described below.

# Science Scienc

First, we used the MCD64A1 product from Collection 5.1, replacing Collection 5 that was used in Randerson et al. (2012). Second, instead of using a single source of Level-3 composited thermal anomaly/fire product from Terra (MOD14A1), here we used individual active fire detections from both Terra and Aqua. Third, to improve geolocation accuracies, we used the MODIS fire location product (MCD14ML) instead of the gridded composite

- fire product (MOD14A1). To further reduce geolocation uncertainties, we only retained active fire detections with small or moderate scan angles (equal or less than 0.5 radians). Even with the above adjustments to improve georegistration, some remaining resampling error was introduced in the process of projecting the variable-size MODIS fire pixels onto the 500 m sinusoidal grid on which the MCD64A1 burned area product is generated. To partially correct this known bias, we applied region-specific factors ranging from 0.88 in northern hemisphere
- Africa to 1.12 for Boreal and Central Asia. These correction factors, which were derived using a rigorous model of the sample-dependent MODIS pixel shape and size, statistically compensated for the simplified, fixed 1-km radius initially used to determine whether an active fire pixel was co-located (inside) or outside of the MCD64A1 burn area pixels. Finally, to estimate dNBR for active fires inside of MCD64A1 burned area, we only used active fire detections for which each of the 4 overlapping 500 m pixels were classified as burned. This
- was a stricter criterion than in Randerson et al. (2012) that increases dNBR<sub>in</sub> and its separation from dNBR<sub>out</sub> and other areas used as controls (Fig. 2).

It was not possible to apply the same constraint in the calculation of  $dNBR_{out}$ , so this adjustment had the effect of lowering  $\gamma$ . At the same time, we raised the filtering standard for control pixels (Eq. 4 of Randerson et al. (2012)) so that pixels within a 1 km buffer area of active fire detections by either Terra or Aqua MODIS were excluded

in the calculation of dNBR for non-burning areas (*dNBR<sub>control</sub>*). During the regional aggregation of dNBR, we excluded 500 m pixels that were marked as 'water' by MODIS land cover type product (MCD12Q1).
 During the time both Terra and Aqua fire detections were available (January 2003-December 2015), we

calculated  $BA_{sf}$  separately for Terra (MOD) and Aqua (MYD).  $BA_{sf}$  was then estimated as the arithmetic mean of the two estimates. A climatological ratio of  $BA_{sf-MYD}/BA_{sf-MOD}$  was used to estimate  $BA_{sf-MYD}$  during periods

when Aqua MODIS observations were not available (August 2000-December 2002). The final GFED4s burned area during the MODIS era was the sum of GFED4 burned area (Sect. 2.3.1; Fig. 1a) and burned area from small fires ( $BA_{sf}$ , Fig. 1b). As expected, burned area from small fires is more prevalent in areas with extensive agriculture and in other human-dominated landscapes (Fig. 1c).

### 30 2.3.3 Estimating burned area prior to the MODIS era (1997-2000) for GFED4s

For the pre-MODIS era, we used monthly active fire data from the Visible and Infrared Scanner (VIRS) aboard the Tropical Rainfall Measuring Mission (TRMM) or the Along Track Scanning Radiometers (ATSR) on board multiple platforms to estimate burned area. Two steps of optimization were used to derive total burned area, starting with the GFED4s product described above. The first step was to develop a relationship between

aggregated active fires (from VIRS or ATSR) and burned area during the MODIS era in each GFED region, with

the aim of using this relationship to estimate regional burned area during 1997-2000. The second step involved distributing the aggregated burned area within each region to individual 0.25° grid cells.

To calculate the regional sum of BA during the pre-MODIS era, we first performed regression analyses between ATSR or VIRS active fires and the regional sum of GFED4s burned area during the MODIS era. We developed

- linear regression models for each GFED region (Fig. 3), for each month, and for each of the five aggregated vegetation classes (see Randerson et al. (2012) for a description of the vegetation classes). When ATSR and VIRS active fire data were both available (Jan 1998-Jul 2000), the highest performing regression from these two datasets was used to estimate the burned area in each region. Among the 14 continental-scale regions, we used VIRS data in Africa, Southeast Asia, Equatorial Asia, and Australia and ATSR data in all other regions (Fig. 4).
- Prior to 1998 when VIRS data were not available, regressions based on ATSR were used. If the ATSR or VIRS active fires for any given month were outside the dynamic range of active fires during the MODIS era, we instead used linear regression derived from all of the monthly data during the MODIS era for that region.

After quantifying the sum of burned area within each region, we distributed it among 0.25° grid cells using the following approach. While active fires from ATSR or VIRS provide some indication about the temporal

- dynamics of fire in a region, the active fire approach tends to underestimate burning in savannas and other areas with herbaceous fuels. To assess how well active fires captured regional spatial patterns, we estimated the spatial correlation between active fires and burned area in each GFED region during the MODIS era. Higher correlations from these analyses indicated better agreement between the spatial distribution of ATSR/VIRS active fires and GFED4s burned area. Since we found the correlation coefficients varied seasonally, a mean
- monthly (*m*) set of spatial correlation coefficients (*SC*) was derived to determine the level of representation of burned area by ATSR/VIRS active fires. The spatial distribution function of burning was based on a linear combination of climatological distribution of burned area (*cl*) and the distribution of active fires (*FC*):

$$BA_{pre-MODIS}(i,t) = BA_{rs}(r,t) \times [SDF_{FC}(r,i,t) \times SC(r,m) + SDF_{cl}(r,i,t) \times (1 - SC(r,m))]$$
(2)

where  $SDF_{FC}$  and  $SDF_{cl}$  are unitless spatial distribution functions that each sum to 1 in each GFED region and were derived from active fire detections or the monthly climatology of burned area during the MODIS-era from GFED4s, and  $BA_{rs}$  is the regional (*r*) sum of burned area for that month and region derived from the regressions between GFED4s and ATSR or VIRS active fires described above. In temperate and high latitude regions, where

- the spatial correlation between active fires and burned area is relatively high, the equation primarily uses information from the pre-MODIS active fires to assign the spatial distribution of burned area. In regions where the spatial correlation between active fires and burned area is relatively low, the equation relies more on the climatological burned area pattern from the MODIS era. For consistency with the previous step, the source of the active fires for generating the SPF was the same as active fires used to generate the regional sum of burned area
- in each region. The contribution of ATSR, VIRS, MCD64A1, and  $BA_{sf}$  to the total burned area is shown in Fig. 4 for the GFED4s time series.

### Searth System Discussion Science Solutions Data

### 2.3.4 Combustion completeness and fire-induced mortality in boreal forests

Despite relatively similar environmental conditions and vegetation attributes, the boreal regions in North America and Eurasia exhibit significantly different patterns of fire severity (Wooster and Zhang, 2004). This was

- shown to primarily be a function of divergent plant traits for the dominant tree species in each continent (Rogers et al., 2015). Species in North America tend to promote crown fires with higher levels of combustion completeness of the canopy and tree mortality compared to lower-severity surface fires in Eurasia. As with other global fire models, GFED3 did not capture these differences due to biome-wide parameterizations. To address the large-scale differences in boreal fire effects, we integrated satellite-based metrics of severity from
- Rogers et al. (2015) including immediate tree mortality and an index of vegetation destruction. These were initially calculated at 1 km and 500 m resolutions, respectively, and aggregated to 1°, but here rescaled to our 0.25° grid without interpolation. Vegetation destruction was derived from three MODIS-based metrics that provide information on immediate fire-induced losses of green vegetation, reduction in canopy and soil water, and landscape charring. These included dNBR, decreases in NDVI, and increases in summer land surface
- temperature (LST). The original vegetation destruction product used LST from Aqua and was available from 2003-2012. We extended it here to 2001 and 2002 using multiple linear regression relationships based on Terra LST, dNBR, and changes in NDVI at 1° ( $r^2 = 0.95$  for North America, 0.96 for Northwest Eurasia, 0.95 for Northeast Eurasia, and 0.91 for Southern Eurasia). Immediate tree mortality was based on decreases in tree cover and increases in spring albedo one year after a fire, and was provided for fires between 2001 and 2009. For both
- products, grid-cell-specific averages were used in years not covered, and grid cells without valid values were assigned regional burned-area weighted means. On average, vegetation destruction was 36 % lower and fire-induced tree mortality was 42 % lower in boreal Eurasia compared to boreal North America. More details on model integration are given in Sect. 3.1, and more information on these products can be found in Rogers et al. (2015).

### 3 Modeling framework and modifications

GFED is based on the Carnegie-Ames-Stanford-Approach (CASA) model that was developed in the early 1990s to simulate the terrestrial carbon cycle using satellite data (Potter et al., 1993; Field et al., 1995; Randerson et al., 1996). In previous work we adjusted the model to account for fires (van der Werf et al., 2003; 2004); further revisions were implemented in GFED2 (van der Werf et al., 2006) and GFED3, including modifications to estimate the contribution of different fire categories including agricultural waste burning, boreal forest fires, deforestation fires, peatland fires, and savanna fires (van der Werf et al., 2010). Below we describe the model in general (3.1), followed by a more detailed explanation of the changes we made in this version (Sect. 3.2-3.5).

### 3.1 CASA-GFED framework

When CASA was developed it computed carbon fluxes as the difference between NPP and  $R_h$ . Both are still calculated for each month and each 0.25° grid cell. NPP is based on a light use efficiency model (Field et al., 1995) and is distributed over various live biomass 'pools' (leaves, stems, roots) according to satellite-derived

5 fractional tree cover maps. In forests we allocate NPP to all three live biomass pools, and in grasslands to leaves and roots, accounting for variability in allocation due to gradients in mean annual precipitation as in GFED3. The carbon in these pools is subsequently delivered to 9 litter pools at the surface and in the soil with turnover rates set for each pool depending on moisture conditions and temperature.

The turnover rates of the wood pool in GFED4 (the modeling framework used to derive both GFED4 and GFED4s emissions) were adjusted at the biome level to match observed aboveground biomass (Avitabile et al., 2016; Santoro et al., 2015). Wood turnover now varies between 40 years for deciduous broadleaf forest and 65 years for deciduous needleleaf forest, with turnover times for evergreen forest in between those values: 52 years for evergreen needleleaf and 55 for evergreen broadleaf (Fig. 5). Similarly, turnover times of slowly-decomposing soil pools were adjusted in GFED4 to better match measured values reported for 0-30 and 30-100 cm (Batjes, 2016).

- In GFED1 we added fire, herbivory, and grazing as additional carbon loss pathways besides  $R_h$ . Fires transfer carbon to the atmosphere and between the different pools depending on the burned fraction of the grid cell, combustion completeness, fire induced mortality rates, and information on whether belowground carbon pools are susceptible to fire or not.
- 20 Combustion completeness (CC) is treated similarly in GFED4 as in our previous work with set minimum and maximum values, see Table 1 in van der Werf et al. (2010). We scaled CC using the soil moisture index (SMI) of the top 7 cm such that the 5<sup>th</sup> and 95<sup>th</sup> percentiles corresponded with the minimum and maximum values. Fire induced tree mortality was set to 2 % for low tree cover regions (mainly savannas and agriculture) and 50 % for forests in general but modified in tropical forests based on fire persistence as in GFED3, and in boreal
- 25 regions according to satellite derived proxy datasets (Sect. 2.3.4). More specifically, in boreal forests we used the satellite-derived instantaneous tree mortality to represent fire-induced tree mortality. In addition, we did not use the CC scaling by SMI for the aboveground wood in the boreal region but used the satellite-derived vegetation destruction scalar for this. The combustion completeness of the wood pool ranged between the set minimum and maximum values (0.2 and 0.4, respectively), and linearly depended on the vegetation destruction scalar instead of SMI.

### 3.2 Modifying the burned fraction to account for sub-grid scale heterogeneity in fuels

In our previous model set-up, fires lowered the fuel load in each grid cell depending on burned area, combustion completeness, and fire induced mortality rates. This was done uniformly in the grid cell not accounting for the fact that fires only lower fuel in the fraction of the grid cell that actually burned. This may have led to an underestimation of emissions in frequently burning regions, especially towards the end of the fire season. For

5

example, in a grassland grid cell that burns in two consecutive months, each with 0.5 burned fraction, modeled fuel loads in the second month are half those of the first month if combustion completeness is set at 100 % (Fig. 6). In reality, the fuel load in that grid cell in the second month should be similar to that in the first month for the part that had not burned, and depleted for the part that had burned. To compensate for this effect we now calculate the *modified burned fraction* of the grid cell as:

$$MBF(i,t) = \frac{\frac{BA(i,t)}{A(i)}}{\left(1 - \frac{\sum_{i=1}^{t-1}BA(i,t)}{A(i)}\right)}$$
(3)

where *MBF* is the modified burned fraction or the modified fraction of the grid cell that burns, *BA* is the burned
area, and *A* is the area of the grid cell at location (*i*). In our hypothetical example from above *MBF* now becomes
1 in the second month according to Eq. 3, thus generating similar emissions in the two months that each burn the same area (Fig. 6). When cumulative burned area over a fire season exceeds the grid cell area this approach yields negative values towards the end of the season; if this occurs these values are replaced by the burned area divided by the grid cell area. Because we only take into account the burned area from the actual month and the
three preceding months, grid cells with two burning seasons are probably not impacted because they usually

occur with more time in between. Our approach does not influence the burned area datasets but only the way it is used in the conversion of burned area to emissions.

### 3.3 Fuel consumption optimization

- Emissions are derived from the multiplication of burned area and fuel consumption per unit burned area, the latter being the product of fuel loads per unit area and combustion completeness. Van Leeuwen et al. (2014) summarized the peer-reviewed literature on fuel consumption rates consisting of 76 studies and covering 121 unique measurement locations. In addition to the fuel consumption measurement, we also included the fuel load measurements mostly in savannas from Scholes et al. (2011) and assumed a combustion completeness of 0.9 for
- these fuel measurements to calculate fuel consumption. This latter set of 95 measurements were mostly confined to South Africa, Botswana, and Zambia.

We used these two compilations to adjust the turnover rates of herbaceous fuels where the largest discrepancies between the model and measurements were found. Uncertainties in the comparison stem from comparing different time period (most measurements were made before our study period) and from comparing local

- measurements with model estimates for 0.25° grid cells. Fuel consumption rates are highly variable, not only between biomes but also within biomes and between separate fuel classes. The overall spatial representativeness of the fuel consumption field measurements is reasonable for most fire-prone regions. However, several important regions from a fire emissions perspective– including Southeast Asia and Central Africa– are underrepresented. For this study we used version 1 of the fuel consumption database available from
- <u>http://www.falw.vu/~gwerf/FC/.</u>

### 3.4 Emission factors

Emission factors are used to convert dry matter burned into emissions of trace gases and aerosols. These were assigned in GFED3 based on the compilation of Andreae and Merlet (2001) with annual updates. A new

- compilation was developed by Akagi et al. (2011) who considered a subset of the available literature focusing on measurements of smoke that had cooled to ambient temperature but had not undergone photochemical processes. In addition to this approach that may better match the requirements from the atmospheric community, Akagi et al. (2011) reported mean values for more biome categories. The most important change in that regard from the GFED perspective is the partitioning of the extratropical forest category into temperate and boreal forests. We
- compiled a subset of the available species that are most frequently used in large-scale chemistry transport models and filled missing values using those of Andreae and Merlet (2001), see http://www.falw.vu/~gwerf/GFED/GFED4/ancill/GFED4 Emission Factors.txt.

### 3.5 Redistributing monthly emissions on daily and 3-hourly timescales

- We made several improvements to the approach described by Mu et al. (2011) for redistributing monthly emissions to daily and 3-hourly time steps in each 0.25° grid cell. This set of higher temporal resolution emissions was created only for the period of 2003 to the present because of increased MODIS active fire data availability after the launch of Aqua.
- To estimate the daily distribution of emissions, we used two sources of information: active fires from 20 MCD14ML and the day of burning reported in the MCD64A1 burned area product. In tropical regions between 25°N and 25°S, we weighted the information content from these two sources equally in grid cells for which both data streams were available. In GFED3, the day of burning was not available for use as a constraint on daily variability. In the extra-tropics (poleward of 25°N and 25°S) we solely used active fires to distribute the daily pattern of emissions. In these regions, gaps between successive overpasses of Aqua and Terra are smaller, and
- active fires have been shown to be moderately effective in capturing daily variations in fire spread rates (Veraverbeke et al., 2014). We removed persistent active fire locations associated with volcanoes, gas flaring, and many other non-fire sources, using a more recent static hotspot database (Randerson et al., 2012). A simple 3-day center mean smoothing filter was applied in tropical regions to adjust for gaps in MODIS coverage, following Mu et al. (2011).
- We created a climatological diurnal cycle of burning in each region and for different aggregated vegetation types to redistribute daily emissions on a 3-hour time step. The approach is similar to the one described in Mu et al. (2011), and uses active fire data derived from full hemispheric scans of GOES-11 (West) and GOES-12 (East) observations during 2007-2009 with version 6.0 of the WF\_ABBA algorithm (Prins et al., 1998; Reid et al., 2009). Here, we used an improved land cover type product from Friedl et al. (2010), MCD12C1 version 5.1,
- during 2007-2009 to create diurnal cycles of emissions for three aggregated vegetation classes within

continental-scale regions in the western hemisphere. These diurnal cycles were then applied in other regions using the same mapping strategy as described in Mu et al. (2011). An example of the redistribution of emissions using this approach for daily and hourly emissions is shown in Fig. 7, showing relatively comparable results as in GFED3.

### 5 4 Results

Over the 1997 – 2015 period, fire emissions according to GFED4s are on average 2.2 Pg C y<sup>-1</sup> with substantial interannual variability. In Sect. 4.1 we discuss the spatial pattern of burned area and the resulting emissions, and in 4.2 the temporal patterns. We then discuss the modeled fuel consumption (4.3) and the greenhouse gas forcing of fires in 4.4. We also explain the main differences between GFED4s and GFED3 as well as differences in

emissions between GFED4s and GFED4, with the latter derived from the same modeling framework but using the burned area dataset without small fires (i.e., with burned area from GFED4) (4.5).

### 4.1 Spatial patterns

The spatial patterns of emissions and burned area are similar but because fuel consumption is, in general, inversely related to fire frequency (Table 1), emissions are less spatially variable than burned area (Fig. 8).

About 84 % of global carbon emissions has an origin in the tropics between 23.5°N and 23.5°S (1830 Tg C y<sup>-1</sup>), and 65% comes from tropical savannas (1418 Tg C y<sup>-1</sup>), underscoring the importance of fire as a driver of biogeochemical cycles and ecosystem processes in tropical ecosystems.

The relative importance of different regions or continents varies depending on whether one is considering burned area, carbon emissions, or trace gas emissions. For example, while Equatorial Asia (mostly Indonesia) is

- responsible for only 0.6 % of global burned area, the region accounts for 8 % of carbon emissions and 23 % of CH<sub>4</sub> emissions from global fire activity. Boreal forests offer a similar, although less extreme, example: 2.5 % of global burned area, 9 % of global fire carbon emissions, and 15 % of global fire CH<sub>4</sub> emissions. This difference is due to the large variability in fire behavior and fuel consumption in forested regions with high fuel loads, especially when fires consume organic soils. The larger contribution of coarse fuels and smoldering stages of
- combustion in organic soils also contributes to higher emission factors for reduced species such as CO and CH<sub>4</sub>. More information on the relative contribution of the different regions is provided in Tables 1 and 2 for fire carbon emissions and on <u>http://www.falw.vu/~gwerf/GFED/GFED4/tables/</u> for individual trace gases and aerosol species.

### 30 4.2 Temporal dynamics

Forest fires are the primary driver of interannual variability in fire emissions (Fig. 9, Table 2). In the tropics, much of this variability is linked with sea surface temperatures, including large-scale climate modes such as El Niño, that alter fire risk in tropical forests (Chen et al., 2016). El Niño years including 1997-1998, 2002, and 2015 have relatively large contributions from tropical forests. Peat burning in Equatorial Asia contribute