# Peer review of "Global fire emissions estimates during 1997-2015"

_Earth System Science Data, 2016_

## Referee Comment (RC1) · Anonymous Referee #1 · 14 Feb 2017

This paper describes the latest version of the Global Fire Emissions Database (GFED). This version, version 4/4s, includes significant updates from GFED3, including updates in the burned area estimates, new fuel consumption parameterizations, more recent input datasets to drive the base calculations of the biogeochemical model that underlies the emissions model, improvements in the way in which small fires are processed, and updated emission factors. The new model updates and results are described in detail and compared to GFED3. While it remains a challenge to prove that these model updates are indeed improvements, as discussed thoughtfully in Section 5, the model updates include significant changes that should be expected to improve emission estimates. The GFED emission estimates are widely used across the climate and atmospheric chemistry communities, and this new version will be very valuable. This manuscript is well-written; the model updates, resulting emission estimates, comparison to past model versions, and discussion of the uncertainties are thorough and clearly stated.

I recommend that this paper be published. I only have minor comments that I provide here.

A link to a file that includes the updated emission factors is provided; however, it would be useful to include this information in the manuscript (A table would be a useful addition).

Section 4.3 (Page 13, lines 16-22): this section is talking about comparisons of measurements with GFED4 and GFED3 modeled fuel consumptions. It is often unclear which is being compared. For example, line 16 states: Fuel consumption in Savannas and other regions with herbaceous fuels is lower in GFED4... lower than measure values? Lower than GFED3? (the next sentence states that it is lower than GFED3 estimates). I just suggest more clearly defining what you are comparing in this section.

Editorial comments: When "which" is used, a comma should preceed it. For example, on page 13, line 30-31 Page 14, line 6: I don't think it's necessary to comment that the "C" emissions are now reported in emissions of CO2. This is confusing.

---

## Referee Comment (RC2) · Anonymous Referee #2 · 6 Mar 2017

This paper documents the development of the Global Fire Emissions Database version 4, largely in relation to previous versions of the dataset. The inevitable assumptions and limitations of the dataset are appropriately described. It is generally a thorough, well-referenced data description paper that can be published as-is, subject to the following minor comments.

P1L30: suggest changing 'This increase' to 'This net change' or 'This net increase'.

P14L15: need a reference for 2014 global fossil fuel emissions in making the comparison to CO2-equivalent fire emissions

P14L21: need a reference for the idea of Savanna fire season management as a climate mitigation instrument

P15L1: suggest changing 'increase' to 'net increase'

---

## Short Comment (SC1) · 6 Mar 2017

The GFED is an important and valuable data product. However, the methodology (Section 2.3.2) to adjust small fire areas (smaller than can be resolved by the 500m MODIS burned area product) may be flawed as it (1) implicitly makes assumptions about the dynamics of natural surfaces pre- and post-fire that may not occur in nature, and (2) ignores non-trivial remote sensing issues concerned with resampling and the scale mismatch between 1km active fire detections and 500m surface reflectance observations.

1) The small fire adjustment allocates burned area to all out-of-burn (i.e., not detected by the 500m MODIS burned area product) 1km MODIS active fire detection pixels. The allocation is undertaken by a multiplicative adjustment (gamma correction factor)

based on the dNBR. The dNBR is defined as the temporal difference in the NBR (a NIR SWIR ratio usually related to burn severity but also to other phenomena such as field tillage intensity) derived from the 16 day 500m MODIS surface NIR and SWIR reflectance stored in the MOD13 VI product.

a) Please quantify the temporal separation between successive NBR values used to compute the dNBR. As written this is unclear but given that the NBR is derived from the 16-day MOD13 VI product that selects a "best" observation in each 16-day period the temporal separation could be up to 30 days apart (or perhaps 45 days apart if the 16-day period in which the MODIS active fire detection occurred was discarded).

b) Please provide a rationale for the validity of the implicit assumption that the surface does not change pre- and post-fire for periods as long as (a). Note, in particular, that there are papers showing that the reflectance and NBR changes rapidly post-fire. What are the implications for this on the small-fires adjustment and where will the adjustment be most prone to departure from this assumption (presumably savannas where surfaces can recover to the pre-burn state in days/weeks, also perhaps over fields that are ploughed, harvested etc.) ?

2) The MODIS is a whiskbroom sensor and the MODIS 1km active fire product detects one, or several, fires that occur anywhere in the pixel footprint that increases in area in the along-track and along-scan directions respectively from approximately ∼1.0 x 1.0 km at nadir to ∼2.0 x 4.8 km at the scan edge. In addition, the MODIS has a triangular response function. See: Wolfe et al. 1998, MODIS Land Data Storage, Gridding, and Compositing Methodology: Level 2 Grid., IEEE Transactions on Geoscience and Remote Sensing, 36, 1324–1338 & Wolfe et al. 2002, Achieving sub-pixel geolocation accuracy in support of MODIS land science, Remote Sensing of Environment, 83, 31-49.

a) Please clarify why a 0.5 radians (28.6 degree) scan angle threshold was used and not some other threshold. What are the along-track and along-scan dimensions of the

MODIS 1km pixel footprint at this angle ? How sensitive are the small fire adjustment results to changing this (arbitrary ?) scan angle threshold.

b) Please explain why the gamma correction factor is not overly sensitive to the large mismatch between the size of the MODIS 1 km active fire footprint and resampled 500m surface reflectance data used to compute the dNBR, including consideration of (i) how small fires can occur anywhere within the 1km active fire footprint, (ii) the MODIS triangular response function.

c) The small fire adjustment builds on the method described in Randerson et al. 2012 (that was not published in a remote sensing journal) and may have similar problems as the above. Please comment if the above issues apply also to the Randerson et al. 2012 paper.

d) Please clarify if an independent, appropriately detailed local to regional scale, verification of the small fire adjustment was undertaken. For example, by comparison with contemporaneous 30m Landsat mapped burned areas (or similar resolution satellite data) in regions where the small fire adjustment resulted in a pronounced change in the total burned area. If not please include such a comparison to reassure the reader that the globally reported 37% increase in burned area (due mostly to the inclusion of small fires) is based on good science.

---

## Author Comment (AC1) · 20 Apr 2017

*We greatly appreciate the comments of the reviewer, please find below our response to the issues raised*

Reviewer: A link to a file that includes the updated emission factors is provided; however, it would be useful to include this information in the manuscript (A table would be a useful addition).

*Response: We have added the Table with emission factors in the revised manuscript*

Reviewer: Section 4.3 (Page 13, lines 16-22): this section is talking about comparisons of measurements with GFED4 and GFED3 modeled fuel consumptions. It is often unclear which is being compared. For example, line 16 states: Fuel consumption in

Savannas and other regions with herbaceous fuels is lower in GFED4. . . lower than measure values? Lower than GFED3? (the next sentence states that it is lower than GFED3 estimates). I just suggest more clearly defining what you are comparing in this section.

*Response: in the revised section we have now clearly outlined whether the comparisons were made against GFED3 or against measurements*

Reviewer: Editorial comments: When "which" is used, a comma should preceed it. For example, on page 13, line 30-31

*Response: We have added a comma wherever "which" introduced a non-restrictive phrase.*

Reviewer: Page 14, line 6: I don't think it's necessary to comment that the "C" emissions are now reported in emissions of $CO_2$. This is confusing.

*Response: This section was written partly to serve the mitigation community where $CO_2$ mass units are much more frequently used than the carbon units. We therefore added the sentence "Note that in this section we refer to C emissions in $CO_2$ mass units rather than the C mass units used in the rest of the paper." This was actually done to prevent confusion and we prefer to keep this sentence in the text.*

---

## Author Comment (AC2) · 20 Apr 2017

*We greatly appreciate the comments of the reviewer, please find below our response to the issues raised*

P1L30: suggest changing 'This increase' to 'This net change' or 'This net increase'.

*Response: changed*

P14L15: need a reference for 2014 global fossil fuel emissions in making the comparison to CO2-equivalent fire emissions

*Response: we have added a reference to Boden et al. (2017)*

P14L21: need a reference for the idea of Savanna fire season management as a climate mitigation instrument

*Response: we have moved the reference to Russell-Smith et al. (2013) from the sentence following the one outline by the reviewer to that first sentence.*

P15L1: suggest changing 'increase' to 'net increase'

*Response: changed*

---

## Author Comment (AC3) · 21 Apr 2017

Response to the short comment written by Dr. David Roy (below we will address his comment in general terms followed by a point-by-point response, our response is in italics)

*We appreciate the comments made by Dr. Roy to ensure that the addition of small fires is based on sound science. It may be good to take a step back and outline the underlying topic for the interested reader not familiar with the difference between "mapped burned area" (on which GFED4 is based) and "small fire burned area" (used for GFED4s, where the small fire burned area is added to the mapped burned area). This, as well as a discussion of the uncertainties in both products was also outlined in the discussion section of our ESSD manuscript.*

*In general, there are two remote sensing products used to monitor fires; burned area and active fires. Both products have advantages and disadvantages, and to date, those two products have been used separately, even though they are complementary; burned area products excel when fires are relatively large, while active fire products can also detect fires that are too small to be picked up by the burned area algorithms. We call the fires not mapped by the burned area algorithms but detected using active fire observations small fires.*

*In principle, if both remote sensing products had perfect geolocation and detection accuracy, all of the active fires should be located within the perimeter of burned areas. In reality, this is not the case, because cloud cover and sub-pixel burns make it challenging to detect burned area associated with small fires in many ecosystems (and, conversely, to detect active fires for all burn scars). The discrepancy between the location of burned areas and the location of active fire detections is not trivial; as shown in Figure 1 of Randerson et al. (2012), the number of active fires outside of 500m burned areas exceeds the number of active fires within or near burn scars for 9 of 14 continental-scale regions, including important burning regions such as temperate North America, Central America, South America, equatorial Asia, and Central Asia. The large number of active fires that do not correspond to burned area pixels highlights the importance and challenge of reconciling these two important data streams.*

*Small fires are important for the biogeochemical and atmospheric communities because they represent a significant amount of total burned area in many ecosystems, and because those fires often occur in relatively densely populated areas where air quality issues are important. Recent literature has shown that GFED (version 3) agreement is not as good with atmospheric observations as the Global Fire Assimilation System (GFAS, Kaiser et al., 2012) based on active fire data in areas where small fires dominate.*

*While there is little doubt that these small fires add to the total burned area, the questions are 1) how much, and 2) whether lowering the omission error in burned area products by adding small fire burned area occurs at the cost of introducing commission errors. Historically, the fire remote sensing community has strived to minimize commission errors. This is critical for effective use of the native resolution products in many applications. Most GFED users, however, are from the atmospheric community better served with a balance between omission and*

*commission errors which is partly enabled by using a much coarser resolution (0.25°) instead of the native 500-meter mapped burned area and 1 km active fire data.*

*We fully agree with Dr. Roy that our approach has considerable imperfections and that in some instances we allocate too much or too little burned area to each active fire observation. At this stage we cannot quantify exactly how much this is but given the improved performance (see Table R1 and Figures S1-S5 below) we feel our new data is one incremental step forward but certainly not the final answer.*

*Using higher resolution data and implementing more complex algorithms (such as longer temporal tracking of surface reflectance) may lead to additional improvements, but those approaches are beyond the scope of our research. Not taking small fires into account leads to flawed results as well. The comparisons shown below in the Table and Figures are for a limited number of regions and while they indicate better results on coarse scales in various biomes, additional evaluation is clearly necessary. When the fire remote sensing community releases their Landsat burned area estimates, increasing the sample size and spatial distribution of validation data for moderate resolution burned area products and small fire estimation methodologies, this will be possible. We anticipate these data will be released later this year, and look forward to working with the remote sensing community on this analysis.*

*The comments of Dr. Roy reminded us that the small fire layer is an interim, experimental dataset. This is now more explicitly mentioned in the main text and we will modify the GFED emissions readme files as well to reflect this. As described in section 2.3.2, we have made several modifications compared to Randerson et al. (2012) that were designed to address some of the limitations of the algorithm, and we intend to adopt a more rigorous approach for next GFED versions, which we will start developing soon also because the GFED4 burned area dataset, which was based on MCD64A1 Collection 5.1, was discontinued after 2016. In this context, in the development of the new GFED products we will carefully consider Dr. Roy's comments and valuable perspective.*

**Point-by-point**

The GFED is an important and valuable data product. However, the methodology (Section 2.3.2) to adjust small fire areas (smaller than can be resolved by the 500m MODIS burned area product) may be flawed as it (1) implicitly makes assumptions about the dynamics of natural surfaces pre- and post-fire that may not occur in nature, and (2) ignores non-trivial remote sensing issues concerned with resampling and the scale mismatch between 1km active fire detections and 500m surface reflectance observations.

*We agree with Dr. Roy that there are potential flaws in our approach. In scientific research, implicit assumptions are often made in order to provide a best guess when the real information is not available or difficult to extract. Our goal was to*

*generate a first-order estimate of the extent of burned area associated with active fires outside the burn perimeters of the 500m MCD64A1 product.*

*We believe our method, which uses currently available moderate resolution global data, improved the quality of a global emissions database. Our approach partially remedies a situation in which omission errors associated with small fires are considerable, and provides an initial estimate of their contribution to regional burned area. On more local scales, our estimation could be improved by using higher-resolution data (e.g., Landsat and Sentinel-2) and implementing more complex algorithms.*

1) The small fire adjustment allocates burned area to all out-of-burn (i.e., not detected by the 500m MODIS burned area product) 1km MODIS active fire detection pixels. The allocation is undertaken by a multiplicative adjustment (gamma correction factor) based on the dNBR. The dNBR is defined as the temporal difference in the NBR (a NIR SWIR ratio usually related to burn severity but also to other phenomena such as field tillage intensity) derived from the 16 day 500m MODIS surface NIR and SWIR reflectance stored in the MOD13 VI product.

*Dr. Roy correctly interprets our use of the difference normalized burn ratio. We chose this index over NDVI or other indices or surface reflectance differences because there is a wide literature documenting the effectiveness of dNBR in mapping fires.*

a) Please quantify the temporal separation between successive NBR values used to compute the dNBR. As written this is unclear but given that the NBR is derived from the 16-day MOD13 VI product that selects a "best" observation in each 16-day period the temporal separation could be up to 30 days apart (or perhaps 45 days apart if the 16-day period in which the MODIS active fire detection occurred was discarded).

*Please see Figure R1 below.*

[Figure]

*Figure R1. NBR separation days for the annual peak burning month in the 14 different GFED regions. Red, orange, and blue represent 'in', 'out', and 'control', respectively. These data were generated from Terra observations during 2012 and indicate a median of 32 days.*

b) Please provide a rationale for the validity of the implicit assumption that the surface does not change pre- and post-fire for periods as long as (a). Note, in particular, that there are papers showing that the reflectance and NBR changes rapidly post-fire. What are the implications for this on the small-fires adjustment and where will the adjustment be most prone to departure from this assumption (presumably savannas where surfaces can recover to the pre-burn state in days/weeks, also perhaps over fields that are ploughed, harvested etc.)

*As can be seen from Figure R1, our method estimates the difference with a median temporal increment of about 32 days. The figure shows that the differences we are computing for active fires within burns (red) have the same distribution of temporal increments as the ones we estimate for active fires outside of burns and for control areas.*

*Generally, the shorter the time between observations the better. However, one can argue for example, that by using 16-day surface reflectance composites one can avoid noise associated with cloud cover or smoke and thus a more reliable signal can be achieved than focusing on shorter periods. Nevertheless, regrowth or other processes including those mentioned by Dr. Roy surely play a larger role in our gamma correction factor than is the case for the change detection component of burned area algorithms that rely on daily surface reflectance imagery, and can confuse the true signal. The very likely implication of this confusion will be commission as well as omission errors at grid-cell level.*

*We note that with a mean temporal increment of about 32 days, the dNBR probability distribution functions show important differences in many regions for active fires outside and inside of burned areas, and these pdfs are different from probability distributions created from non-burning control areas. These distributions are shown in Figure 2 in the main text, and the separability of the different pdfs has improved relative to those shown in Randerson et al. (2012) as a consequence of improvements in the algorithm noted in the main text.*

*We note that the probability distribution of dNBR from unburned controlled areas, sampled with the same temporal distribution, is essential for estimation of gamma in our algorithm (equation 4 of Randerson et al. (2012)). Our approach implicitly assumes that the pdfs of dNBR sampled from the active fires inside burned areas and outside burned areas temporally evolve in the same way within each of the three seasonal compositing intervals.*

2) The MODIS is a whiskbroom sensor and the MODIS 1km active fire product detects one, or several, fires that occur anywhere in the pixel footprint that increases in area in the along-track and along-scan directions respectively from approximately ~1.0 x 1.0 km at nadir to ~2.0 x 4.8 km at the scan edge. In addition, the MODIS has a triangular response function. See: Wolfe et al. 1998, MODIS Land Data Storage, Gridding, and Compositing Methodology: Level 2 Grid., IEEE Transactions on Geoscience and Remote Sensing, 36, 1324–1338 & Wolfe et al. 2002, Achieving sub-pixel geolocation accuracy in support of MODIS land science, Remote Sensing of Environment, 83, 31- 49.

a) Please clarify why a 0.5 radians (28.6 degree) scan angle threshold was used and not some other threshold. What are the along-track and along-scan dimensions of the MODIS 1km pixel footprint at this angle? How sensitive are the small fire adjustment results to changing this (arbitrary ?) scan angle threshold.

*The scan angle threshold of 0.5 radians corresponds to along-scan dimension of ~1.37 km and along-track dimension of ~1.16 km. So the pixel size at this scan angle is ~1.6 km². We used this value to limit the area of view on one hand, and to include enough sample size on the other hand. We note the selection of pixels closer to nadir reduces co-registration errors between surface reflectance imagery and active fire locations, and that is why we introduced this improvement here. It was not included in the original algorithm described by Randerson et al. (2012). Detailed sensitivity tests are needed in a future study. We added the following sentence to the relevant section: "The threshold was somewhat arbitrary and future research is required to identify how a balance between sample size and area of view is best achieved."*

b) Please explain why the gamma correction factor is not overly sensitive to the large mismatch between the size of the MODIS 1 km active fire footprint and resampled 500m surface reflectance data used to compute the dNBR, including consideration of (i) how small fires can occur anywhere within the 1km active fire footprint, (ii) the MODIS triangular response function.

*An analysis we conducted since publication of Randerson et al. (2012) demonstrates that the gamma correction factors are indeed highly sensitive to the unavoidable resampling process involved in 1) generating the gridded surface reflectance imagery used in our approach and 2) co-registering 1-km MODIS swath pixels onto the 463-m resampled sinusoidal grid. The net result is that our small-fire burned area estimates have an imprint of this resampling error superimposed upon them. Despite this limitation, our estimates will still be useful to the atmospheric community since the pressing concern within this community is the abundance of unreported small-fire burned area. Importantly, the small fire method only adds burned area where out-of-burn fire pixels are present, thus even if the magnitude of the small-fire burned area is highly uncertain, that area is assigned to the proper location and day of burning.*

c) The small fire adjustment builds on the method described in Randerson et al. 2012 (that was not published in a remote sensing journal) and may have similar problems as the above. Please comment if the above issues apply also to the Randerson et al. 2012 paper.

*The issues above apply to Randerson et al. (2012) as well. In the current manuscript, we have made several improvements to the algorithm described in that paper. These include 1) reducing the scan angle in the estimation of inside and outside active fire populations to reduce registration errors, 2) the use of both Terra and Aqua data to better sample different aspects of the diurnal cycle, and 3)*

*more stringent controls on the development of the dNBR pdf of active fires within burn perimeters. Together, these represent an improvement from Randerson et al. (2012), but we believe new advances in this area must come from extensive use of higher spatial resolution imagery which is also mentioned in the manuscript.*

d) Please clarify if an independent, appropriately detailed local to regional scale, verification of the small fire adjustment was undertaken. For example, by comparison with contemporaneous 30m Landsat mapped burned areas (or similar resolution satellite data) in regions where the small fire adjustment resulted in a pronounced change in the total burned area. If not please include such a comparison to reassure the reader that the globally reported 37% increase in burned area (due mostly to the inclusion of small fires) is based on good science.

*As in Randerson et al. (2012), we have conducted comparisons in several regions, including Alaska, Canada, the contiguous U.S., Mali, and Portugal. As expected, the algorithm makes only a minor adjustment in regions such as Alaska and Canada where out-of-burn fire pixels are comparatively rare (Fig. R2, panels a and b). Importantly, the algorithm approximately doubles burned area for prescribed and agricultural fires in the U.S., and brings our estimates into much closer agreement with independent estimates of prescribed fires (Fig. R2, panel d) and alternative estimates of agricultural fires (Fig. R2, panel c). Comparisons with higher resolution observations from Mali (Table R1) and Portugal (Figs. R3-R5) also provide confidence that our approach will generally not overestimate the burned area contributed by small fires.*

*In conclusion, we recognize the valid concerns of Dr. Roy and acknowledge the limitations of our small fire estimates. We hope Dr. Roy and the larger remote sensing community also appreciate that many researchers are better served with burned area estimates that better balance omission and commission errors.*

*In this context, it is essential that future validation work quantify burned area biases across different biomes and continents and seasonal periods. In a recent paper by Padilla et al. (2015) published in RSE, for example, the authors report low biases in burned area of 44% for the collection 5 MCD64A1 product and 48% for MCD45A1 from an analysis of 103 randomly distributed Landsat scenes. While these moderate resolution burned area products had the best performance of the 6 products that that the authors evaluated, these biases are considerable and provide motivation for the type of adjustments we implemented here.*

*Understanding the small fire burned area contribution has become an important research focus, and clearly the interim, dNBR-based approach we have used must be replaced in the future. We have highlighted in the abstract, discussion, and conclusions of the revised version that the small fire layer is uncertain and that over the next years those estimates will be revised ("This small fire layer carries substantial uncertainties; improving these estimates will require use of new burned area products derived from high-resolution satellite imagery.").*

[Figure]

*Fig R2. Comparison of MODIS burned area products with regional burned area products for North America. Updated from Figure 5 of Randerson et al. (2012). BA TOTAL (with small fires) shown in blue is the same as the 0.25° product described in van der Werf et al. (2017) as GFED4s. For panels a-b, the observations (black lines) are from large fire GIS perimeter datasets. These perimeters often encapsulate unburned islands within complex patterns of landscape burning. Thus, it is expected that both MCD64A1 and GFED4s burned area estimates will be lower, because of the ability of the MODIS 500m observations to more accurately resolve (and exclude) unburned islands.*

*Table R1. Burned area (MCD64A1 and GFED4s) comparison with Landsat-derived estimates from Laris (2005). Updated from Table 2 of Randerson et al (2012).*

| Time period | $BA_{MCD64A1}$ (% of study area per month or year) | $BA_{GFED4s}$ (% of study area per month or year) | Landsat-derived burn area (% of study area per month or year) |
|---|---|---|---|
| September-November 2002 | 1.1 | 5.0 | 17.6 |
| December 2002 | 12.2 | 13.7 | 23.9 |
| January 2003 | 4.6 | 6.7 | 10.5 |
| February 2003 | 1.7 | 2.9 | 4.9 |
| Total 2002-2003 fire season | 19.6 | 28.3 | 56.9 |

[Figure]

*Fig R3. Burned area for Portugal derived from Landsat imagery. Updated from Figure S3 of Randerson et al (2012). BA total shown in blue is the same as the 0.25° product described in van der Werf et al. (2017) as GFED4s.*

[Figure]

*Figure R4. Annual mean burned area from Portugal during 2002 ‐ 2010 compared to the MODIS burned area products. Updated from Figure S4 of Randerson et al. (2012). BA total is the same as the 0.25° product described in van der Werf et al. (2017) as GFED4s.*

[Figure]

*Figure R5. Burned area from a national polygon fire dataset from Portugal during 2001-2010 (observed BF) compared to the MODIS burned area products. Updated from Figure S5 of Randerson et al (2012). The solid circles represent the 0.25⁰ product described in van der Werf et al. (2017) as GFED4s. The open circles represent GFED4 burned area derived from MCD64A1.*

**References**

Kaiser, J. W., Heil, A., Andreae, M. O., Benedetti, A., Chubarova, N., Jones, L., Morcrette, J.-J., Razinger, M., Schultz, M. G., Suttie, M., and van der Werf, G. R.: Biomass burning emissions estimated with a global fire assimilation system based on observed fire radiative power, Biogeosciences, 9, 527-554, doi:10.5194/bg-9-527-2012, 2012.

Laris, P. S.: Spatiotemporal problems with detecting and mapping mosaic fire regimes with coarse-resolution satellite data in savanna environments, Remote Sens. Environ., 99(4), 412–424, doi:10.1016/j.rse.2005.09.012, 2005.

Padilla, M., Stehman, S. V., Ramo, R., Corti, D., Hantson, S., Oliva, P., Alonso-Canas, I., Bradley, A. V., Tansey, K. J., Mota, B., Pereira, J. M., and Chuvieco, E.: Comparing the accuracies of re- mote sensing global burned area products using stratified random sampling and estimation, Remote Sens. Environ., 160, 114–121, doi:10.1016/j.rse.2015.01.005, 2015.

Randerson, J. T., Chen, Y., Werf, G. R., Rogers, B. M., and Morton, D.: Global burned area and biomass burning emissions from small fires, J. Geophys. Res., 117, G04012, doi:10.1029/2012JG002128, 2012.

van der Werf, G. R., Randerson, J. T., Giglio, L., van Leeuwen, T. T., Chen, Y., Rogers, B. M., Mu, M., van Marle, M. J. E., Morton, D. C., Collatz, G. J., Yokelson, R. J., and Kasibhatla, P. S.: Global fire emissions estimates during 1997–2015, Earth Syst. Sci. Data Discuss., https://doi.org/10.5194/essd-2016-62, in review, 2017.